

# Outcomes of critically ill end-stage kidney disease patients who underwent major surgery

Peerawitch Petchmak[1], Yuthapong Wongmahisorn[1] and Konlawij Trongtrakul[2,3]

[1] Department of Surgery, Faculty of Medicine Vajira Hospital, Navamindradhiraj University, Bangkok, Thailand
[2] Department of Medicine, Faculty of Medicine Vajira Hospital, Navamindradhiraj University, Bangkok, Thailand
[3] Department of Medicine, Faculty of Medicine, Chiang Mai University, Chiang Mai, Thailand

## ABSTRACT

**Purpose**. End-stage kidney disease (ESKD) is a major worldwide health problem. Patients with ESKD are thought to have a significant risk for development of complications following an operation. However, the study of ESKD and its outcomes following major operations remains rare, particularly in critical illness. Therefore, this study aimed to demonstrate how the outcomes of ESKD patients were affected when they underwent a major operation and were admitted to the intensive care unit (ICU), compared with non-ESKD patients.

**Methods**. A retrospective matched case cohort study was conducted in 122 critically ill surgical patients who underwent a major operation and were admitted to the ICU, during 2013 and 2016. Sixty-one ESKD patients who required long-term dialysis were enrolled and compared with 61 matched non-ESKD patients. The matching criteria were the same age interval ($\pm$5 years), gender, and type of operation. The ICU mortality was compared to the primary outcome of the study.

**Results**. Patients' baseline characteristics between ESKD and non-ESKD were similar to a priori matching criteria and other demographics, except for pre-existing diabetes mellitus and hypertension, which were found significantly more in ESKD ($p = 0.03$ and 0.04, respectively). For operations, ESKD showed a higher grade of the American Society of Anesthesiologist (ASA) physical status ($p < 0.001$), but there were no differences for emergency surgery ($p = 0.71$) and duration of operation ($p = 0.34$). At ICU admission, the severity of illness measured by the Sequential Organ Failure Assessment (SOFA) score was greater in ESKD ($8.9 \pm 2.6$ vs $5.6 \pm 2.5$; $p < 0.001$). However, after eliminating renal domain, SOFA non-renal score was equivalent ($5.7 \pm 2.2$ vs $5.2 \pm 2.3$, $p = 0.16$). The ICU mortality was significantly higher in critically-ill surgical patients with ESKD than non-ESKD (23% vs 5%, $p=0.007$), along with hospital mortality rates (34% vs 10%, $p = 0.002$). The multivariable logistic regression analyses adjusted for age and SOFA non-renal score demonstrated that ESKD had a significant association with ICU and hospital mortality (adjOR = 5.59; 95%CI [1.49–20.88], $p = 0.01$ and adjOR = 4.55; 95%CI[1.67–12.44], $p = 0.003$, respectively).

**Conclusion**. Patients who underwent a major operation and needed intensive care admission with pre-existing ESKD requiring long-term dialysis were associated with greater mortality than patients without ESKD. More careful assessment before, during,

Corresponding author
Konlawij Trongtrakul,
konlawij@live.com

and after major surgical procedures should be performed in this group of patients to improve post-operative outcomes.

## INTRODUCTION

Chronic kidney disease (CKD) is a major worldwide public health problem (*Couser et al., 2011*; *Hill et al., 2016*; *Ong-Ajyooth et al., 2009*). Patients suffering from CKD may experience uremia, anemia, cardiovascular diseases, and decreased quality of life (*Couser et al., 2011*; *Jha et al., 2013*). They can also progress to long-term dialysis, kidney transplantation, or even death, when they proceed to end stage kidney disease (ESKD).

The development of ESKD has been increasingly reported over the past decades (*Collins et al., 2014*; *Thammatacharee & Suphanchaimat, 2020*). In the United States, data suggests that the incidences of ESKD are projected to increase up to 11–18% by the year 2030, compared to those of 2015 (*McCullough et al., 2019*). This feature could be explained by the current situation of a growing aging population (*Goulding, Rogers & Smith, 2003*). Moreover, the predispositions of patients toward ESKD have been increasing; including obesity (*Hales et al., 2018*), hypertension (*Forouzanfar et al., 2017*), and diabetes mellitus (*Geiss et al., 2014*). Patients with ESKD are more likely to have complications with cardiovascular diseases, gastrointestinal bleeding, malnutrition, or immunosuppression (*Bagshaw & Uchino, 2009*; *Chan & Ostermann, 2013*; *Sood et al., 2011*; *Uchino et al., 2003*). These complications mostly require hospitalization and create economic burdens (*Praditpornsilpa et al., 2011*). An increase of incidences (1% to 1996 to 3% in 2010) of dialysis-requiring acute kidney injury (AKI) in critically-ill surgical patients is found (*Wald et al., 2015*) and is also likely to be found in critically-ill patients with pre-existing ESKD. Previous studies have reported the incidence of intensive care unit (ICU) admissions of ESKD at around 1–11% (*Apel et al., 2013*; *Bagshaw & Uchino, 2009*; *Hutchison et al., 2007*; *Jha et al., 2013*; *Strijack et al., 2009*). In addition, this population has a greater chance of admission to the ICU (25 times) and is associated with a higher mortality rate than the general population (*Hutchison et al., 2007*; *Jha et al., 2013*; *Strijack et al., 2009*).

Critically-ill patients with ESKD consume more ICU resources than those who were not ESKD (*Arulkumaran, Annear & Singer, 2013*; *Fidalgo & Bagshaw, 2014*), particularly, the challenges of the maintenance of regular dialysis, the proper management of fluids, and the regulation of metabolism and electrolytes. Moreover, the EKSD patients have several negative impacts including longer length of ICU stay and longer length of hospitalization (*Fidalgo & Bagshaw, 2014*) and may be confronted with several long-term effects following intensive care discharge, for instance, cardiovascular complications, malnutrition, and deconditioning (*Fidalgo & Bagshaw, 2014*).

This trend of increasing ESKD patients and resulting complications highlights the need for improved resource utilization systems within a limited resource environment, such as the surgical ICU. Numerous barriers have been found to preclude proper ICU bed allocation (*Van Sluisveld et al., 2017*), particularly regarding emergency surgery (*Hasan, Bahalkeh & Yih, 2020*). Inevitably, ESKD patients will in some cases acquire diseases requiring surgical treatment and be at risk of becoming critically-ill. Surgical ICU is a specialized unit serving patients who undergo general surgery related to acute care and life-threatening conditions. Care of critically-ill surgical patient with ESKD needs to be further understood in relation to their physiology being disrupted by surgical intervention. For example, is there a greater risk of morbidity and mortality among critically-ill ESKD patients who have undergone major operations, even though current evidence about the surgical outcomes is lacking. A new study is needed to address these concerns.

Three factors contributed to the development of this study: the possibility that surgical interventions may induce stressors to ESKD patients differently from medical problems, data related to this study have mostly been drawn is from mixed medical and surgical intensive care units (ICU), and includes limited patient profiles regarding type of operation and perioperative information. The purpose of the present study, therefore, is to explain how the outcomes of critically-ill ESKD patients who undergo major operations are affected when compared with surgical critically-ill non-ESKD patients. It is hypothesized that critically-ill ESKD patients who undergo major operations will be independently associated with a greater risk of mortality than surgical critically-ill non-ESKD patients.

## MATERIALS & METHODS

The study was approved by the Institutional Review Board of the Faculty of Medicine Vajira Hospital, Navamindradhiraj University, Bangkok, Thailand (approval no. 088/57). Waiving of informed consent was allowed due to the minimal risk of the study and its retrospective nature. We conducted a retrospective matched cohort study in critically-ill surgical patients with and without ESKD by searching for cases who had undergone a major operation and had been admitted to our three surgical ICUs (a total of 17 beds) during the period of January, 2013 through December, 2016.

Inclusion criteria are adult critically-ill surgical patients (age equal to 18 years and above) with and without ESKD who had undergone a major operation and had been admitted to the surgical ICU within the first 48 h post-operatively. Patients transferred to the ICU due to serious medical conditions not related to any operation, or critically-ill surgical patients who underwent specific procedures related to CKD/ESKD, such as arteriovenous fistula, dialysis vascular access or kidney transplantation, were excluded from the study.

To reduce selection bias, critically-ill surgical patients without a history of ESKD were matched to critically-ill ESKD patients a priori with the same three baseline characteristics. The matching characteristics included age interval ($\pm$ 5 years), gender, and type of operation, their significance in determining higher surgical risk. Previous data from critically-ill patients undergoing major operations show that age (older than 65), is independently associated with mortality (*Elia et al., 2013*). Gender, one of the general

baseline demographics, has been found as a covariate of high mortality for females (*Romo, Amaral & Vincent, 2004*). In addition, greater severity of illness at surgical ICU admission, as measured by the SOFA score is associated with higher mortality rates (*Pornwaragron et al., 2019*).

## Definitions

Critically-ill ESKD patients were defined as those requiring chronic dialysis of at least 6 weeks before ICU admission by either hemodialysis (HD) or peritoneal dialysis (PD) (*Apel et al., 2013*; *Strijack et al., 2009*). Some critically-ill surgical patients without ESKD might have pre-existing renal dysfunction, so kidney disease staging was defined according to KDIGO-2012 criteria (Kidney Disease: Improving Global Outcomes (KDIGO) CKD Work Group. 2013). Major operation was defined according to all procedures that required either general or regional anesthesia. Major operations were done based on surgeon decision and standard practice guidelines in the study center.

## Data collection

Baseline characteristics including patient age, gender, body weight, height, body mass index (BMI), and pre-existing comorbidities were extracted from electronic medical records. Pre-existing comorbidities including the presence of diabetes mellitus, hypertension, dyslipidemia, cardiovascular disease, cerebrovascular accident, and chronic kidney disease were recorded. The histories of ESKD patients were reviewed for their mode of long-term dialysis, which included HD and PD.

Details about the methods of anesthesia, including general or regional anesthesia, were collected, together with the patients' American Society of Anesthesiologists (ASA) physical status. In addition, we collected the operative data regarding whether surgery was emergency or elective, type of operation, and duration of operation.

Illness severity at ICU admission was calculated by Sequential Organ Failure Assessment (SOFA) score. However, to eliminate the effect of serum creatinine levels, which might be greater in ESKD patients, we omitted the renal domain of the SOFA score and named it SOFA non-renal score. In addition, comorbidity during ICU admission was collected including sepsis and septic shock (pneumonia, surgical site infection, etc.), cardiovascular disease (acute coronary syndrome, congestive heart failure, and cardiac arrhythmia), gastrointestinal bleeding, cerebrovascular accident, deep vein thrombosis, and electrolyte imbalance (such as hypokalemia, hyponatremia, hypomagnesemia).

The primary outcome of the study was ICU mortality. Hospital mortality, ICU co-morbidities, and complications during the ICU stay, ICU length of stay, and hospital length of stay were reported as secondary outcomes.

## Sample size

The sample size to enhance statistical power in our study was calculated according to the study by *Apel et al. (2013)*. They reported 23.1% of ESKD critically-ill surgical patients died during the ICU stay, while 5.5% were found to be non-EKSD patients. Based on these rates, we assumed a sample size of 122 (61 cases for each group) was needed to demonstrate the effect of ESKD on critically-ill surgical patient outcomes. Critically-ill surgical patients

with ESKD were randomly selected based on available retrospective data. In the non-ESKD group, patients were selected according to the above-mentioned matching characteristics.

## Statistical analysis

Continuous variables were summarized as mean ($\mu$) and standard deviation (SD) for normally distributed data, or median and interquartile range 1 and 3 (IQR1-3) for non-normal distributed data. Categorical descriptive variables were summarized as frequency ($n$) and percentages (%).

Statistical analysis plan for comparing mean and medians were between the two groups using Student's $t$-test or Mann–Whitney U test, when appropriate. Categorical variables were analyzed with Fisher Exact test. We performed multivariable logistic regression analysis calculating Odds ratios (OR) and its 95% confidence interval (95%CI) for identifying how ESKD affected the primary outcome and adjusted by covariates. The overparameterization of the multivariable regression model is a concern, due to a limited sample size in this study. Generally, it is accepted that events per variable should be greater than 10 when generating a multivariable regression model (*Deng et al., 2017*; *Steyerberg, Schemper & Harrell, 2011*). If the number of deaths is 30 out of a total of 122 cases (25%), no more than 3 variables are allowed to be included in the model. Moreover, the most commonly used criteria for the variable selection in the multivariable regression model is $p$-value of less than 0.10–0.20 in the univariable analysis (*Deng et al., 2017*; *Harris et al., 2015*; *Trongtrakul et al., 2019*). However, a priori variable selection is accepted (*Walter & Tiemeier, 2009*). This study used significant variables that are associated with a greater mortality risk in surgical ICU, including age (*Elia et al., 2013*) and severity of illness measured by SOFA non-renal score (*Pornwaragron et al., 2019*) as adjusted variables to understand the association between ESKD and mortality. The STATA statistical software version 13.0 (StataCorp LP, College Station, TX, USA) was used for statistical analyses and $p$-value of less than 0.05 was considered statistically significant.

## RESULTS

### Baseline characteristics of the study

A total of 122 surgical critically-ill patients were included in the study, 61 critically-ill with ESKD and 61 critically-ill without ESKD (Fig. 1). Fifty-five ESKD patients were commenced on long-term HD, while another 6 patients were commenced on PD. All three matching characteristics were comparable between ESKD and non-ESKD. They were similar in age (67.5 ± 10.1 years vs 66.8 ± 10.6 years, $p = 0.42$), the proportion of males (57% vs 59%, $p = 0.85$), and types of operation (all $p > 0.05$) (Table 1).

Most pre-existing comorbidities, comprising of dyslipidemia, cardiovascular disease, and cerebrovascular diseases, were similar between groups, except diabetes mellitus and hypertension, which were more prevalent in ESKD than non-ESKD (67% vs 46%, $p = 0.03$ and 89% vs 72%, $p = 0.04$, respectively). Moreover, at least 7/61 (12%) of non-ESKD patients had known baseline renal dysfunction categorized as CKD stage 2-4.

For operations, ESKD patients had a higher grade of ASA classification ($p < 0.001$), but ESKD had no significant effect on the technique selected for anesthetizing ($p = 0.49$) or

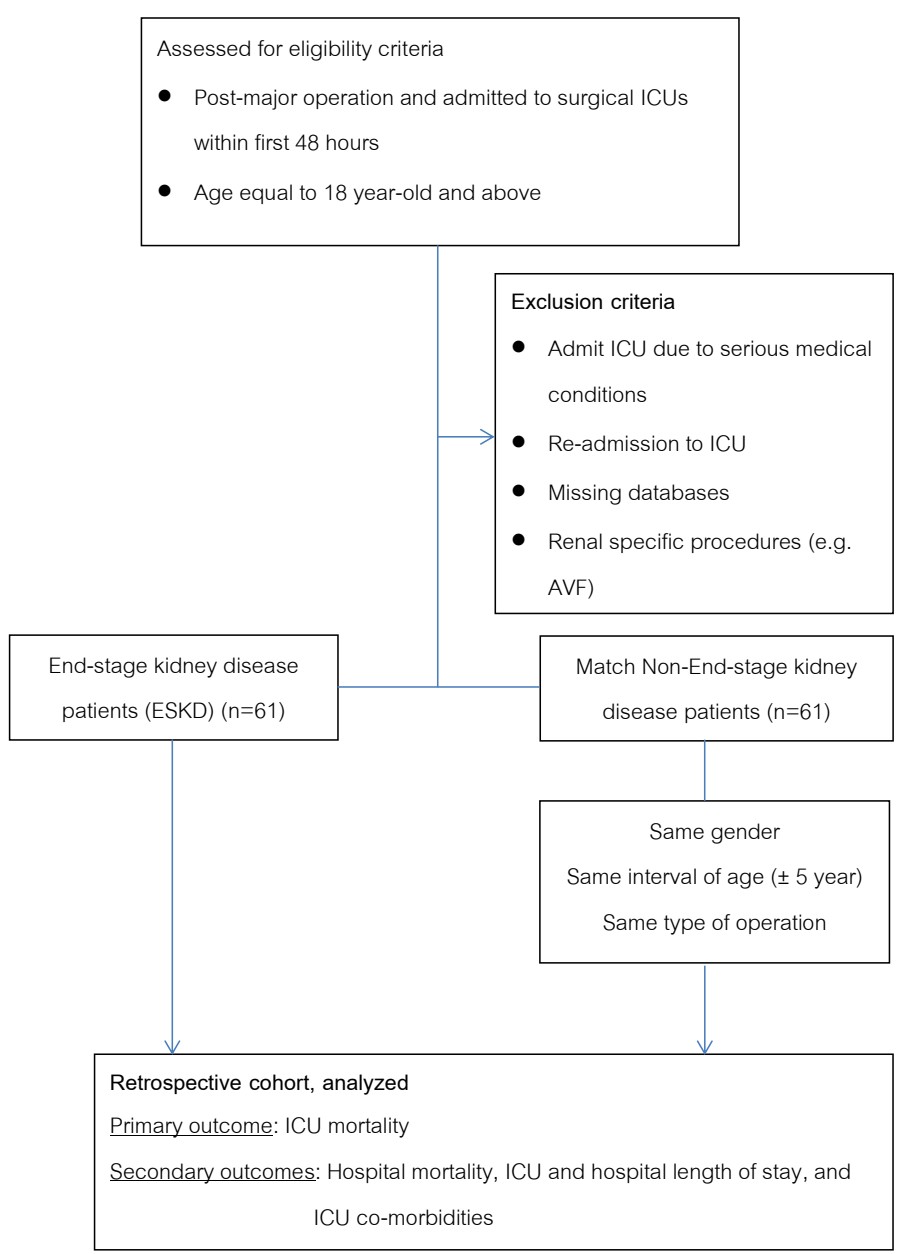

**Figure 1** **Flow of the study.**

on the status of emergency surgery ($p = 0.71$). The majority of operations were vascular related surgery (excluding arterio-venous graft or catheterization for long term dialysis), followed by cardiothoracic and gastrointestinal tract surgery. Between the two groups, we did not find any differences in the types of operation (all $p > 0.05$) and duration of operation (4 h [IQR 2,6 hours] vs 3.5 h [IQR 2,5 hours], $p = 0.34$).

At the time of ICU admission, the severity of illness measured by SOFA score was higher in ESKD than non-ESKD ($8.9 \pm 2.6$ vs $5.6 \pm 2.5$, $p < 0.001$). However, after eliminating

**Table 1  Baseline characteristics of the study group on admission to the intensive care unit according to the presence of end-stage kidney disease (ESKD) and non-ESKD.**

| Characteristics | ESKD ( $n = 61$) | Non-ESKD ( $n = 61$) | P-value |
|---|---|---|---|
| Age - years | 67.5 ±10.1 | 66.8 ±10.6 | 0.71 |
| Male - $n$ (%) | 35 (57%) | 36 (59%) | >0.99 |
| Body weight - kg | 58.5 ±11.4 | 60.7 ±11.7 | 0.30 |
| Height - cm | 161 ±8 | 160 ±8 | 0.26 |
| Body mass index - kg/sq-m | 23.7 ±3.8 | 22.6 ±4.8 | 0.16 |
| Pre-existing conditions, $n$ (%) | | | |
| Diabetes mellitus | 41 (67%) | 28 (46%) | 0.03 |
| Hypertension | 54 (89%) | 44 (72%) | 0.04 |
| Hyperlipidemia | 20 (33%) | 22 (36%) | 0.85 |
| Cardiovascular disease | 23 (38%) | 20 (33%) | 0.71 |
| Cerebrovascular accident | 10 (16%) | 6 (10%) | 0.42 |
| Chronic kidney disease | – | 7 (12%) | – |
| End stage kidney disease | 61 (100%) | – | – |
| ASA classification, $n$ (%) | | | <0.001 |
| 2 | 0 (0%) | 12 (20%) | |
| 3 | 52 (85%) | 45 (74%) | |
| 4 | 9 (15%) | 4 (6%) | |
| Anesthesia technique, $n$ (%) | | | 0.49 |
| General anesthesia | 55 (90%) | 58 (95%) | |
| Spinal block | 6 (10%) | 3 (5%) | |
| Emergency surgery, $n$ (%) | 26 (43%) | 23 (38%) | 0.71 |
| Types of surgery, $n$ (%) | | | |
| Gastrointestinal surgery | 8 (13%) | 17 (28%) | 0.07 |
| Vascular surgery | 21 (34%) | 11 (18%) | 0.06 |
| Urology | 7 (12%) | 7 (11%) | >0.99 |
| Hepato-pancreato-biliary | 2 (3%) | 2 (3%) | >0.99 |
| Neurosurgery | 1 (2%) | 1 (2%) | >0.99 |
| Cardiovascular thoracic | 13 (21%) | 14 (23%) | >0.99 |
| Others | 9 (15%) | 9 (15%) | >0.99 |
| Operative time (hours)[*] | 4 (2,6) | 3.5 (2,5) | 0.34 |
| Illness severity at ICU admission | | | |
| SOFA | 8.9 ±2.6 | 5.6 ±2.5 | <0.001 |
| SOFA non renal | 5.7 ±2.2 | 5.2 ±2.3 | 0.16 |

**Notes.**
Reported as median and Interquartile range 1 and 3; ASA, the American Society of Anesthesiologist; ICU, intensive care unit; SOFA score, the Sequential Organ Failure Assessment score.

the calculation of serum creatinine from the renal domain of SOFA score, both groups had a non-significant difference in organ dysfunction (SOFA non-renal score = 5.7 ± 2.2 vs 5.2 ± 2.3, $p = 0.16$).

## Primary and secondary outcomes

ICU mortality rates were significantly greater in ESKD than non-ESKD (23% vs 5%, respectively; $p = 0.007$). ESKD also affected hospital mortality (34% vs 10%, $p = 0.002$)

**Table 2   Morbidity and mortality of the study group according to the presence of end-stage kidney disease.**

| Outcomes | ESKD ( $n=61$ ) | Non-ESKD ( $n=61$ ) | *P*-value |
|---|---|---|---|
| Primary outcome | | | |
| ICU mortality, *n* (%) | 14 (23%) | 3 (5%) | 0.007 |
| Secondary outcomes | | | |
| Hospital mortality, *n* (%) | 21 (34%) | 6 (10%) | 0.002 |
| ICU length of stay - days* | 3 (2,12) | 2 (1,6) | 0.007 |
| Hospital length of stay - days* | 29 (16,51) | 20 (14,34) | 0.03 |
| ICU co-morbidities, n (%) | | | |
| Severe sepsis/septic shock | 24 (39%) | 23 (38%) | >0.99 |
| Pneumonia | 11 (18%) | 9 (15%) | 0.81 |
| Surgical site infection | 7 (11%) | 14 (23%) | 0.22 |
| Cardiovascular diseases | 19 (31%) | 15 (25%) | 0.55 |
| Acute coronary syndrome | 11 (18%) | 4 (7%) | 0.10 |
| Congestive heart failure | 7 (11%) | 7 (11%) | >0.99 |
| Cardiac arrhythmia | 4 (7%) | 8 (13%) | 0.36 |
| Gastrointestinal bleeding | 2 (3%) | 3 (5%) | >0.99 |
| Cerebrovascular accident | 3 (5%) | 1 (2%) | 0.62 |
| Deep venous thrombosis | 1 (2%) | 0 (0%) | >0.99 |
| Electrolytes imbalance | 9 (15%) | 12 (20%) | 0.63 |

**Notes.**
Reported as median and interquartile range 1 and 3; ICU, intensive care unit.

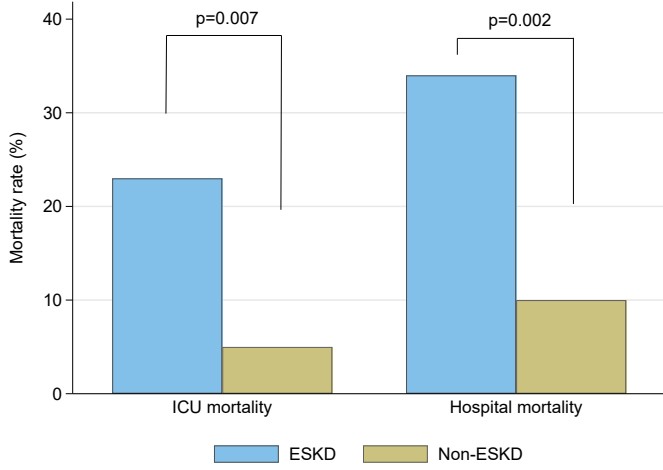

**Figure 2   Mortality rates between critically-ill surgical patients with ESKD and non-ESKD populations.**

as shown in Table 2 and Fig. 2. When comparing the modalities for long-term dialysis in

**Table 3** Multivariable regression analyses the association of ESKD affected on ICU mortality and hospital mortality.

|  | OR | 95%CI | *P* value | AdjOR[*] | 95%CI | *P* value |
|---|---|---|---|---|---|---|
| ICU mortality | 5.76 | 1.56 to 21.24 | 0.009 | 5.59 | 1.49–20.88 | 0.01 |
| Hospital mortality | 4.81 | 1.78 to 13.01 | 0.002 | 4.55 | 1.67–12.44 | 0.003 |

Notes.
[*]Adjusted by age and SOFA non-renal score.

ESKD, there was no difference in risk for ICU mortality between HD and PD (14/55 [25%] cases vs 0/6 [0%], respectively, $p = 0.19$).

The median time of ICU length of stay was significantly longer in ESKD than non-ESKD (3 days [IQR 2,12 days] vs 2 days [IQR 1,6 days], respectively, $p = 0.007$). Moreover, longer hospital length of stay was also found (29 days [IQR 16,51 days] vs 20 days [IQR 14,34 days], respectively, $p = 0.03$).

Table 2 also illustrates our patients' comorbidities during admission to the ICU. The presence of sepsis and septic shock was the most common, followed by cardiovascular diseases, 39% (47/122 cases), and 28% (34/122 cases), respectively. Despite many complications or co-morbidities during the ICU stay, there were no differences between ESKD and non-ESKD (all $p \geq 0.05$).

## ESKD and risk of mortality

Table 3 shows the association between ESKD and its effect on outcomes. A greater risk of ICU mortality was found (OR = 5.76; 95%CI [1.56–21.24], $p = 0.009$). In addition, hospital mortality dominated in ESKD groups (OR = 4.81; 95%CI [1.78–13.01], $p = 0.002$). After adjusting for age and SOFA non-renal score, both ORs remained statistically significant (OR = 5.59; 95%CI [1.49–20.88], $p = 0.01$ and OR = 4.55; 95%CI [1.67–12.44], $p = 0.003$, respectively).

# DISCUSSION

## Summary of the study

The main finding of our study demonstrated that critically-ill surgical patients suffering long-term dialysis from ESKD who experienced a major operation had around a 5-time greater risk of ICU mortality and hospital mortality than critically-ill surgical patients without ESKD. Moreover, ESKD also had an impact on longer duration of ICU stay and hospital stay than non-ESKD.

## Comparing to the previous study

To the best of our knowledge, data regarding the effect of ESKD on critically-ill surgical patients' outcomes have scarcely been reported. An extensive search revealed the largest cohort of surgical intensive care patients from the study by *Apel et al. (2013)*. The study specifically investigated surgical intensive care patients who underwent operation (same features as ours) and found an ESKD prevalence of 1.5% of almost 13,000 critically-ill surgical patients. The results showed that patients with ESKD had higher ICU mortality and hospital mortality than non-ESKD (23% vs 6%, $p < 0.001$ and 31% vs 10%, $p < 0.001$,

respectively). These rates were quite similar to ours (23% vs 5%, $p = 0.007$ and 34% vs 10%, $p = 0.002$, respectively). Moreover, the ICU length of stay was similar between the Apel M, et al. study (2 days [IQR 1,7 days] vs 1 days [IQR 1,3 days], $p < 0.001$) (*Apel et al., 2013*) and ours (3 days [IQR 2,12 days] vs 2 days [IQR 1,6 days], $p = 0.007$). Controversy regarding ESKD's association with poorer outcome in critically-ill patients was reported. Most of the studies confirmed ESKD as a predictor of poor prognosis for mortality among the critically-ill (*Hutchison et al., 2007*; *Manhes et al., 2005*; *Sood et al., 2011*), although some studies contradicted this (*Strijack et al., 2009*; *Uchino et al., 2003*). However, all of them studied from mixed medical and surgical critically-ill patients, which might explain why outcomes were inconsistent.

As mention above, only *Apel et al. (2013)* studied ESKD and surgical outcomes among the critically-ill. In their multivariable analyses, ESKD was independently associated with a greater risk of hospital mortality (adjOR = 3.84, 95%CI [2.68–5.50], $p < 0.001$ when adjusted for age, gender, co-morbidities, SAPS II, type of surgery, and SOFA nonrenal score). In our study we identified quite similar features; ESKD was associated with both ICU and hospital mortality (adjOR = 5.59; 95%CI [1.49–20.88], $p = 0.01$ and adjOR = 4.55; 95%CI [1.67–12.44], $p = 0.003$, respectively; in ours, age and SOFA non-renal score were used for adjusting).

Our study was unique because we reported information about anesthetic and perioperative data, which have rarely been reported. A trend for a higher grade of ASA physical status was found in our study. This may be explained by patients with ESKD are thought to be incapable and therefore expose themselves to greater risks from operative procedures than general surgical patients. The three most common surgical procedures in our study were vascular, cardiothoracic, and gastrointestinal surgery. Similarly, *Apel et al. (2013)* reported cardiothoracic surgery as being the most common procedure. This corresponded to ESKD patients always having cardiovascular problems due to comorbid diabetes mellitus and hypertension, which are most generally known as common causes of ESKD (*Couser et al., 2011*; *Ong-Ajyooth et al., 2009*).

We found that sepsis and cardiovascular diseases were the first two common causes of our ICU comorbidities, a finding noted in previous studies (*Hutchison et al., 2007*; *Manhes et al., 2005*; *Strijack et al., 2009*). This means for patients already on long-term dialysis, more vital strategies should be focused upon, such as good pre-operative preparation, more invasive hemodynamic monitoring, and optimal fluid therapy, among others.

In addition, due to a scarcity of evidence in critically-ill surgical patients with ESKD, the mortality rates and its outcomes of ESKD and those of AKI were examined. The ICU mortality from AKI in critically-ill surgical patients was greater in AKI than those non-AKI groups (26% vs 3%, $p < 0.001$) (*Trongtrakul et al., 2019*) and a greater hospital mortality (19% vs 4%; $p = 0.0001$) (*Harris et al., 2015*). Moreover, length of stay is longer when comparing the AKI to the non-AKI groups. The studies from *Trongtrakul et al. ( 2019)* reported longer median days in the ICU and the hospital (6 days [IQR 3,13 days] vs 1 days [ IQR 1,3 days], $p < 0.001$ and 18 days [IQR 10, 28 days] vs 14 days [IQR 9,24 days], $p = 0.003$, respectively). Longer median days was also demonstrated in *Harris et al. (2015)* (6 days [IQR 3-10 days] vs 3 days [IQR 2-5 days], $p = 0.001$ and 19 days [IQR 10-30

days] vs 9 days [IQR 5-16 days], $p = 0.0001$, respectively). Although there is a difference in the population studied, both the ESKD and the AKI groups when undergoing a major operation and encountering a critical illness show a greater chance for mortality and longer length of stay in the ICU and in the hospital. Determining the different impact from ESKD and AKI in the critically-ill surgical patients is important for future studies.

## Strengths

Due to a lack of information regarding ESKD's effect on critically-ill surgical patients' outcomes, our study adds more information to this gap in knowledge. ESKD alone also has higher morbidity and mortality than the general population. In combination with major surgery, a known cause of overwhelming stress, this could lead to greater morbidity and mortality than among those with general surgical critical illness. The severity of illness, measured by SOFA score, included serum creatinine level or urine output; in ESKD they are uncommonly low at baseline. Therefore, SOFA non-renal score was also reported as one of the baseline characteristics. A priori matching reduced differences between ESKD and non-ESKD participants. One might argue that propensity score matching or a larger observational study should be done for exploring the hypothesis; however, it was not possible in our setting, due to a lack of technology to support large data collection. All patient data was manually extracted from the scanned medical records under our best clinical practice and statistical methodology.

## Limitations

There were some limitations in our study. Some information regarding ESKD was not collected, for instance, the prevalence of ESKD in our cohort and pre-ICU admission data about ESKD (e.g., dialysis vintage, dialysis adequacy, residual renal function). The other issue was a limitation in multivariable logistic regression analysis. We would have liked to adjust confounders that had an association with mortality. However, the low number of mortality outcomes limits the number of variables included in the model. We are aware of overparameterization when we included variables with more than one predictor per ten-event outcomes. Therefore, the model was adjusted by a common demographic when patients were admitted to the ICU, including age and SOFA non-renal score. Heterogeneity of type of operation might be another interference on the outcome in our study. This should be investigated in a larger scale study to identify pre-, peri-, and post-operative risk of mortality in surgical critically-ill patients who underwent major operations. This could be useful in pre-operative evaluation and management to improve this group of patient outcomes. Moreover, ESKD patients, who exert a greater demand for ICU admission and a higher chance of mortality, should be fully evaluated for the factors that have an impact on surgical ICU performance, for better and more appropriate resource allocation and management.

## CONCLUSION

Critically-ill surgical patients with ESKD were strongly associated with higher mortality compared to non-ESKD patients. Besides appropriate surgical wound and drainage care,

invasive monitoring to understand the complexity of the patient volume status and scheduling adequate dialysis to maintenance patient metabolic and electrolyte homeostasis are encouraged. Moreover, to achieve the most favorable outcome for critically-ill surgical patients with ESKD who undergo a major operation, acute care must be endorsed throughout the pre-, peri-, and post-operative period.

## ACKNOWLEDGEMENTS

We would like to acknowledge the Faculty of Medicine Vajira Hospital, Navamindradhiraj University for providing information. We also would like to kindly thank Mr. Jason D. Cullen and Mr. Eric B. Tedstrom, MSW who helped to review and edit our manuscript.

### Funding

The authors received no funding for this work. However, the fee for publication was supported by the research fund of the Faculty of Medicine Vajira Hospital and the research fund of Navamindradhiraj University, Bangkok, Thailand. The funders had no role in study design, data collection and analysis, decision to publish, or preparation of the manuscript.

### Grant Disclosures

The following grant information was disclosed by the authors:
Faculty of Medicine Vajira Hospital.
Navamindradhiraj University, Bangkok, Thailand.

### Competing Interests

The authors declare there are no competing interests.

### Author Contributions

- Peerawitch Petchmak and Konlawij Trongtrakul conceived and designed the experiments, performed the experiments, analyzed the data, prepared figures and/or tables, authored or reviewed drafts of the paper, and approved the final draft.
- Yuthapong Wongmahisorn conceived and designed the experiments, analyzed the data, authored or reviewed drafts of the paper, and approved the final draft.

### Human Ethics

The following information was supplied relating to ethical approvals (i.e., approving body and any reference numbers):

The Institutional Review Board of the Faculty of Medicine, Vajira Hospital granted ethical approval to carry out the study (approval no. 088/57).

### Data Availability

The raw data are available in the Supplementary File.

## Supplemental Information

Supplemental information for this article can be found online at http://dx.doi.org/10.7717/peerj.11324#supplemental-information.

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
