# Peer review of "Outcomes of critically ill end-stage kidney disease patients who underwent major surgery"

_PeerJ, doi:10.7717/peerj.11324_

## Round 0.1 · original submission · Minor Revisions

This paper makes an interesting and novel contribution to the literature on the survival of patients treated for ESKD and is richly deserving of publication. The authors are to be congratulated for their well designed and adequately powered research work.

You should note and respond to the useful comments about your statistical methods made by one of the reviewers. Also, it is essential to get some input from a native English speaker as while your meaning is quite clear, some of the grammar and syntax you have used is not yet at a publishable standard.

Reviewer 1 ·

Basic reporting

Some English language issues will likely be cleared up in the editing process.

A few of note:

1) line 16- 1%11% should be 1-11%

2) line 35- "We conducted a retrospective matched cohort" should be "We conducted a retrospective matched cohort study"

3) line 46- "criteria" should be "characteristics"

Experimental design

1) mention in the abstract and in line 92 that "multivariable" logistic regression analaysis was performed.


2) The authors state they performed multivariate analysis with only a few variables (age, SOFA-renal score, ESKD) to avoid "overparameterization". However, it is standard practice in these types of studies to perform univariate logistic regression with all relevant variables, then identify those that are significant or approach significance (usually with P value <0.1) and include them in multivariate analysis. It would be preferable if the authors could do this. If not, I accept their explanation of "overparameterization", but this should also be mentioned in the methods section when discussing the multivariate logistic regression analysis.

3) I would classify the ICU comorbidities as post-surgical outcomes and include them as secondary outcomes, rather that baseline characteristics. As such, I think they would benefit from being in a separate table instead of included in Table 1.

Validity of the findings

1) in the abstract, the sentences "Patients baseline characteristics between the ESKD and non-ESKD were similar as in priori matching criteria, which were all p>0.05). Pre-existing diabetes mellitus and hypertension were found more in ESKD significantly (p=0.03 and 0.04, respectively)" contradict each other. It should be mention that baseline characteristics were similar "except for" diabetes and hypertension.

2) It would be worthwhile for the authors to analyse and discuss the causes of mortality in the ESKD and non-ESKD cohorts and if there was a difference herein. I can see that the cohorts did not differ in terms of incidence of comorbidities during their ICU stay. Therefore, which comorbidities actually led to the increased mortality of ESKD patients?

Reviewer 2 ·

Basic reporting

Discussion section needs to be revised. Also, the overall paper can benefit from English editing. See my comments for the details.

Experimental design

There are several minor concerns regarding the choice of matching criteria, as well as outcome variables. See my comments.

Validity of the findings

Findings can be further validated using similar works from the literature. I mentioned a few areas for improvement in my comments.

Additional comments

See the attachment.

Annotated reviews are not available for download in order to protect the identity of reviewers who chose to remain anonymous.

Reviewer 3 ·

Basic reporting

This is a retrospective study conducted with ESRD patients underwent major operation in single institute. The manuscript is well written, but still lack of something need to elucidate. 1. As the title mentioned, the study groups were ESRD and no-ESRD, however in the ESRD group in post-operative ICU care, what did the patients receive dialysis therapy, still HD or CVVH?
2.In the comparison group, ie, no-ESRD patients, was there any AKI, acute kidney injury, and need further transient treatment to stablize their hemodynamic, fluid status, intake/output...?
3. What were the cause of death in this study? And it is better to present in tables to offer more details.

Experimental design

well

Validity of the findings

This is a retrospective study conducted with ESRD patients underwent major operation in single institute. The manuscript is well written, but still lack of something need to elucidate. 1. As the title mentioned, the study groups were ESRD and no-ESRD, however in the ESRD group in post-operative ICU care, what did the patients receive dialysis therapy, still HD or CVVH?
2.In the comparison group, ie, no-ESRD patients, was there any AKI, acute kidney injury, and need further transient treatment to stablize their hemodynamic, fluid status, intake/output...?
3. What were the cause of death in this study? And it is better to present in tables to offer more details.

Additional comments

This is a retrospective study conducted with ESRD patients underwent major operation in single institute. The manuscript is well written, but still lack of something need to elucidate. 1. As the title mentioned, the study groups were ESRD and no-ESRD, however in the ESRD group in post-operative ICU care, what did the patients receive dialysis therapy, still HD or CVVH?
2.In the comparison group, ie, no-ESRD patients, was there any AKI, acute kidney injury, and need further transient treatment to stablize their hemodynamic, fluid status, intake/output...?
3. What were the cause of death in this study? And it is better to present in tables to offer more details.

---

## Round 0.2 · Major Revisions

Although reviewer #1 made no further comments on the manuscript, there are issues raised by reviewer 2 that deserve full consideration. As you will gather from the comments, reviewer #2 noted that the some issues raised initially were not addressed sufficiently and also made specific comments on the study methodology and robustness of the findings. All issues raised must be addressed thoroughly and unequivocally in the revised version.

Reviewer 1 ·

Basic reporting

No further issues

Experimental design

No further issues

Validity of the findings

No further issues

Reviewer 2 ·

Basic reporting

Insufficient context and not enough motivations based on existing gaps. This concern was raised initially and unfortunately not being addressed sufficiently.

Experimental design

Choices of variables not being supported from the literature. Also, experiment details are lacking as being mentioned in the first round.

Validity of the findings

Since majority of key variables (based on the literature and other studies) are missing, I would be concerned about generalizing the outcomes and findings.

---

## Round 0.3 · accepted · Accept

All issues raised during the review process were addressed satisfactorily

Reviewer 1 ·

Basic reporting

All my previous comments have been satisfactorily addressed.

Experimental design

All my previous comments have been satisfactorily addressed.

Validity of the findings

All my previous comments have been satisfactorily addressed.